# Adequacy of Disease Control by Supraomohyoid Neck Dissection in cT1/T2 Tongue Cancer

**DOI:** 10.3390/jpm12091535

**Published:** 2022-09-19

**Authors:** Andrea Iandelli, Filippo Marchi, An-Chieh Chen, Chi-Kuan Young, Chun-Ta Liao, Chung-Kan Tsao, Chung-Jan Kang, Hung-Ming Wang, Tung-Chieh Joseph Chang, Shiang-Fu Huang

**Affiliations:** 1Department of Otorhinolaryngology and Head and Neck Surgery, Chang Gung Memorial Hospital, Linkou Branch, Yaoyuan 33302, Taiwan; 2Unit of Otorhinolaryngology-Head and Neck Surgery, IRCCS Ospedale Policlinico San Martino, 16132 Genova, Italy; 3Department of Surgical Sciences and Integrated Diagnostics (DISC), University of Genova, 16132 Genova, Italy; 4Department of Plastic Surgery, Chang Gung Memorial, Linkou Branch, Taoyuan 333, Taiwan; 5Department of Otolaryngology, Head and Neck Surgery, Chang Gung Memorial, Keelung Branch, Keelung 20401, Taiwan; 6School of Medicine, Chang Gung University, Taoyuan 333, Taiwan; 7Department of Medical Oncology, Internal Medicine, Chang Gung Memorial, Linkou Branch, Taoyuan 333, Taiwan; 8Department of Radiation Oncology, Chang Gung Memorial, Linkou Branch, Taoyuan 333, Taiwan; 9Graduate Institute of Clinical Medical Sciences, Chang Gun Medical College, Chang Gung University, Taoyuan 33302, Taiwan

**Keywords:** oral cancer, tongue cancers, head and neck cancer, neck dissection, supraomohyoid neck dissection, skip metastases

## Abstract

Background: Patients affected by oral tongue squamous cell carcinoma (OTSCC) underwent a supraomohyoid neck dissection (SOHND) or modified radical neck dissection (mRND) according to the clinical nodal status (cN0 vs. cN+). We investigate whether the type of neck dissection affects survival with the presence of extranodal extension (ENE) and multiple nodal metastases (MNM). Methods: We conducted a retrospective study enrolling surgically treated patients affected by cT1/T2 OTSCC and MNM or ENE. The outcomes assessed were: overall survival (OS), disease-free survival (DFS), and neck-control- and metastases-free survival (NC-MFS). Survival curves were plotted by the Kaplan–Meier method and the log-rank test. Furthermore, we conducted a multivariable analysis with the Cox regression model. Results: We included a total of 565 patients (36% cT1, 64% cT2). Of these, 501 patients underwent a SOHND, and 64 underwent an mRND. A total of 184 patients presented rpN+, with 28.7% of these in the SOHND group and 62.5% of these in the mRND group. We identified no significant differences in OS, DFS, and NC-MFS in the whole pN+ cohort, in the MNM, and the ENE subgroups. In the multivariable analysis, the type of ND did not affect OS and DFS. Conclusions: Treating cT1-2 N0/+ tongue cancer with SOHND is oncologically safe. ENE and MNM patients do not benefit from an mRND.

## 1. Introduction

Oral cavity squamous cell carcinoma (OSCC) contributes substantially to the global cancer burden, with relevant cancer-specific mortality in the region of 31% in most major referral centers [1,2,3,4]. More than fifty-two thousand cases are diagnosed annually in the United States [5]. The tongue is one of the most common subsites involved, found in up to half of all cases [6], with a propensity for early and extensive local invasion and regional lymphatic spread, leading to significant disease- and treatment-related morbidity. Regional neck metastasis represents an ominous prognostic factor in OSCC. In oral tongue squamous cell carcinoma (OTSCC), the presence of just one metastatic lymph node (LN) commits patients to an advanced-stage disease category [7]. It has been shown to determine up to a 50% decrease in overall survival (OS) [8,9,10,11,12]. Moreover, OTSCC has a higher propensity for lymphovascular invasion (LVI), perineural invasion (PNI), and neck dissemination than other OSCC subsites, one of the most significant prognostic factors and varies from 6% to 85% [13,14,15,16,17]. Thus, historically modified radical neck dissection (mRND) has been part of the treatment of OTSCC. However, the cosmetic and functional defects associated with mRND prompted the search for less morbid alternatives [18,19,20].

Detailed examination of lymphatic drainage pathways showed that levels I–II were at the most significant risk of metastasis in OSCC, with levels IV and V playing minor roles. Consequently, to overcome these issues and minimize morbidities in the past decades, elective selective neck dissection (SND) was described to clear the nodal levels with a higher chance of occult metastases [21]. Afterward, several studies demonstrated that in OTSCC, the most commonly nodal levels at risk of metastasis are levels I, II, and III [4,13,22]; therefore, supraomohyoid neck dissection (SOHND) resulted in achieving comparable oncological outcomes to mRND [23,24]. The impact of selective neck dissection in patients with clinically evident nodal disease (cN+) is more controversial. Recent studies [25,26] reported reasonable regional control with selective neck dissection in selected patients with cervical nodal metastases, mainly when adjuvant treatment was applied. Nevertheless, several questions need to be answered: which subpopulation of patients would benefit more from a less extensive dissection? How often are skip metastases, defined as nodal metastases that bypass levels I, II, or both, localized directly to levels III, IV, or V [27], found in early-stage OTSCC? This study aimed to investigate whether, in a group of clinically early-stage OTSCC with the presence of detrimental risk factors, such as extranodal extension (ENE) and multiple nodal metastases, an mRND might determine a survival advantage.

## 2. Materials and Methods

### 2.1. Patients Selection

We conducted a retrospective analysis of all patients affected by clinically early-stage (cT1-T2) cancer of the mobile tongue treated at our institute between March 1997 and May 2017. The institutional review board approved the study at Chang Gung Medical Foundation, Taiwan (IRB No. 202101807B0). The inclusion criteria were: age over 18 years old; a biopsy-proven squamous cell carcinoma (SCC) of the mobile tongue; at least one year of follow-up, or an earlier date of recurrence or death; clinical stage (cT1-T2) according to AJCC 8th edition [7]; treated with curative intent with upfront surgery including neck dissection as part of the treatment. In addition, we excluded patients with a previous history of head and neck cancer; treatment with neoadjuvant therapy; advanced or very advanced local stage; unavailable follow-up information; prior chemotherapy for any cancer and/or radiotherapy in the head and neck area; synchronous head and neck squamous cell carcinoma; presence of distant metastases.

### 2.2. Diagnostic Work-Up, Treatment Policy, and Follow-Up

All patients received a complete tumor survey, including a detailed medical history, comprehensive physical examination, blood tests, computed tomography (CT) or magnetic resonance imaging (MRI) of the head and neck, chest radiographs, abdominal sonography, bone scan, or positron emission tomography (PET). Tumor excision was performed according to the lesion’s extent with a macroscopic surgical margin of at least 1 cm. Free flap reconstruction was performed when needed. Patients without clinically suspicious lymph node metastases (cN0) underwent supraomohyoid neck dissection (SOHND). A modified radical neck dissection (mRND) was performed in the presence of clinically evident nodal disease (cN+). Adjuvant treatment was discussed by the multidisciplinary head and neck tumor board, including radiotherapy or chemoradiotherapy [28,29]. Clinical data recorded included gender, age, smoking, and treatment details. Pathological details recorded were surgical margin status, diameter, adverse pathological features, pN category, and nodal metastases. All tumors were re-staged according to the 8th edition of the UICC-AJCC Staging System [7]. Follow-up included clinical examination every two months during the first and second year, every three months in the third year, every four months in the fourth year, and every six months during the fifth year. No routine radiological follow-up was performed unless patients had symptoms or suspicious clinical finding(s) of recurrence for which CT/PET was carried out.

### 2.3. Survival Analysis

The x2 test evaluated differences and relationships between categorical parameters. Survival analysis was performed considering as outcomes the overall survival (OS), defined as the time between the date of treatment and the date of death; disease-free survival (DFS), defined as the time between the date of treatment and recurrence; and neck control and metastases-free survival (NC-MFS), as defined as a regional or distant relapse of the disease. We evaluated the impact of the different types of neck dissections on the aforementioned oncological outcomes. We first assessed the whole cohort of patients; then, we further analyzed the subgroups of patients affected by multiple nodal disease and extranodal extension (ENE) because these are the subcategory believed to benefit from more extended neck treatment. Survival curves were plotted by the Kaplan–Meier method, and the log-rank test compared the differences between curves. To reduce the effect of confounder values in OS and DFS, with conducted a multivariable analysis with the Cox regression model. Statistical analyses were performed by SPSS 18.0 software (SPSS Inc., Chicago, IL, USA), and *p* < 0.05 was considered statistically significant. The survival curves were constructed with packages of “survival” and “survminer” in R language version 4.1.0 (R Foundation for Statistical Computing, Vienna, Austria) [30,31,32,33].

## 3. Results

### 3.1. Patients, Disease, and Treatment Characteristics

We collected data from 583 patients affected by cT1/T2 OTSCC treated at our institution between March 1997 and May 2017. After a careful assessment, we excluded 18 patients due to missing data, identifying a total number of 565 patients amenable to the study. The clinical characteristics of the patients are presented in Appendix A. The mean age of the study group was 49.73 years old; it was mainly composed of males (85.8%). Among this cohort, 256 patients (45.3%) were initially classified as cT1 and 309 (54.7%) as cT2. After the pathology report’s results and evaluation of the depth of invasion, 175 were staged as pT1 (31%), 199 (35.3%) as pT2, and 191 (33.8%) as pT3, according to the AJCC 8th edition staging system. After preoperative staging, 501 patients were considered negative for regional metastasis (cN0) and therefore underwent an elective SOHND, whereas 64 patients showed a suspicious pathological lymph nodes disease (cN+); consequently, according to the department’s policy, an mRND was performed. The pathologic characteristics of the cohort are listed in Table 1. The occult nodal metastases (cN0/pN+) rate was 28.7% in the SOHND group. In the mRND group, nodal metastases were confirmed in 62.5% of cases, with 184 patients affected by regional disease. Regarding the site of pathologically confirmed metastasis (Appendix A), level I was the most involved (58.1%), followed by level II (55.9%) and level III (26.6%); level IV and V were affected in a small percentage of cases: 9.3% and 1.5%, respectively. No skip metastases were detected. In the SOHND group, 71.3% of the patients were pN0, 11% pN1, 5.6% pN2, and 12.2% were affected by pN3b disease. In the mRND group, the composition was: pN0 37.5%, pN1 6.2%, pN2 4.7% and pN3b 51.6%. Patients that received mRND had a higher rate of pN3b with a statistically significant difference compared to the SOHND cohort (*p* < 0.001). Results are summarized in Table 2.

We further analyzed the subgroups of patients with multiple nodal metastases (Table 3) and ENE disease. These are the subcategory believed to benefit from more extensive neck treatment. In addition, we identified a different ENE distribution in patients who underwent mRND, with a higher rate of extracapsular spread than the SOHND group, 89.3% and 61.1%, respectively (*p* < 0.001).

### 3.2. Survival Analysis by Type on Neck Dissection

Considering the OS as the endpoint, in the pathologic positive lymph nodes group, the 3 years OS rate was 63% for the SOHND and 47% for the mRND; we did not observe a statistically significant difference between the two (*p* = 0.119) (Figure 1). Moreover, no difference was revealed in regional relapse and distant failure rate (*p* = 0.509) (Figure 1). The NC-MFS curves did not show survival advantages in the mRND group compared to the SOHND group.

In the cohort of patients affected by multiple nodal diseases, the 3 years OS rate was 58% in the SOHND group and 43% in the mRND group (Figure 2). There was no significant difference (*p* = 0.105) in the overall death rate between the two treatments. Furthermore, no statistically significant different trend (*p* = 0.523) was observed in DFS and NC-MFS. (Figure 2). Lastly, regarding the subgroup of patients with ENE’s presence, the 3 years OS was 54% in patients subjected to SOHND and 42% in patients who underwent mRND. No statistically significant differences were identified affecting any of the outcomes assessed: OS (*p* = 0.592), DFS (*p* = 0.444), and NC-MFS (Figure 3).

The associations of clinicopathologic factors, including age, pathologic T stage, pathologic N stage, the type of ND, histologic grade, and PNI status with DFS and OS, are summarized in Table 4. At the multivariable analysis, the risk factors that significantly affected the DFS were: a positive lymph node (pN+ENE−), (HR: 1.603; 95% C.I.: 1.085–2.369, *p* = 0.018), and the presence of PNI (HR 1.398, 95% C.I.: 1.023–1.910, *p* = 0.035), whereas an age ≥ 50 had a protective effect (*p* = 0.043). Furthermore, the combination of pN+ with ENE increased the risk of recurrence b twofold (HR: 2.034, 95% C.I.: 1.381–2.995; *p* ≤ 0.001).

Analyzing the OS, the variables determining a negative impact on outcome were an increase in pT stage: pT2 (HR: 2.245, 95% C.I.: 1.310–3.848, *p* = 0.003) and pT3 (HR: 2.910, 95% C.I.:1.691–5.007, *p* < 0.001), respectively; furthermore, the presence of pN+ ENE(−) and (pN(+) ENE(+), HR: 3.756; 95% C.I.: 2.510–5.769; *p* < 0.001) were confirmed to represent a detrimental prognosticator likewise in OS. On the other hand, the covariable “type of neck dissection” showed no significant effect on any of the survival outcomes studied.

## 4. Discussion

The indications and the extent of neck dissection in early-stage tongue cancer have historically been a matter of debate for different reasons, arguing the risk–benefit evaluation between the probability of neck metastases, complications, and the prognostic influence of delayed diagnosis of metastasis during follow-up [23]. Unfortunately, retrospective studies on cN0 cT1/T2 OTSCC did not help solve this problem; moreover, in cN+, the rationale between the extent of the dissection and improvement of survival is not well defined yet [34]. With this study, we report the clinical outcomes of cT1/T2 OTSCC following a retrospective pathology review of specimens according to AJCC 8th edition guidelines [7]; evaluate the impact of SOHND versus mRND in pN0 and pN+; and, in the case of ENE, we evaluate the percentage of skip metastasis.

Decades after Crile first described the radical neck dissection, shoulder and neck pain following the procedure was viewed as a minor and acceptable side effect [35]. Even the initial description of the disability after radical neck dissection in 1951 [36] characterized postoperative discomfort as “variable and seldom incapacitating”; they noted that most patients “continue without much difficulty to their living as before surgery” [36]. However, a systematic review of the literature reported that the prevalence of shoulder pain and dysfunction after mRND was markedly higher compared with SND (range, 9–25%). Moreover, mRND was associated with a reduced health-related quality of life [37]. Therefore, surgeons nowadays are motivated to modify the classic radical neck dissection to preserve function and improve life’s quality [20] while achieving equivalent oncologic effectiveness [38,39]. Thus, over the past 50 years, there have been sequential developments in modified, functional, and selective neck dissections [40]. Moreover, surgeons kept paying more attention to mini-invasive approaches, hidden-scar incisions, and postoperative rehabilitation [35,41].

Concerning oral tongue cancer, regional lymph node metastasis incidence varies from 6% to 85% [11,13,15]. Since then, the need for lymph node clearance in cT1-2N0 tongue cancers has been a matter of much debate. Finally, in a randomized controlled trial from 2015, D’Cruz et al. demonstrated that END in patients with early-stage oral carcinoma improves overall and disease-free survival compared to observation [42]. They reported a rate of 30% of occult nodal metastasis (false negative cN0/pN+), in line with our result of 28.7% and ones from previous literature [22], reaching 35% in cases of floor-of-mouth involvement [43]. The following year, a metanalysis confirmed that END significantly reduces the regional failure rate and improves DSS in patients with cT1-2N0 oral cancer [44].

An up-to-date meta-analysis, including 19 studies, reported that the occult nodal metastasis rate ranges from 10.2% to 44.6% (mean 24.4%) [16]. The authors, analyzing T1 and T2 separately, found that the occult metastasis rate was not significantly different from the studies with more T2 cases than T1, which means that the nodal spread does not depend only on the T stage; thus, elective neck treatment might be needed even in minor cases.

A milestone study carried out in India [42] and those that followed [16,44] demonstrated that all cT1/T2 OTSCC need to receive elective neck dissection since 25–30% is an unacceptable rate of subclinical neck metastasis to avoid neck clearance, despite T stage. However, we decided to enroll only patients affected by early clinical stage (cT1–T2), given that, in similar studies, an advanced stage demonstrated an influence on the DSS and OS, regardless of the type of neck dissection [45].

Nevertheless, the presence of N+ is one of the more significant prognosticators [11]. Thus, elective mRND has been part of treating patients with cN+ squamous cell carcinoma of the oral tongue [46,47]; ideally, it should be avoided in patients with clinically false-positive and true-negative lymph nodes. We noted that 37.5% of patients that received MRND for cN+ disease turned to be pN0 (false positive), meaning that more than one-third of the cohort were overtreated and, instead, could have undergone SOHND obtaining an equal cure rate (the covariable “type of neck dissection” was demonstrated to not significantly affect any of the survival outcomes assessed). Similarly, Iype et al. reported no significant difference in regional recurrence between pN0 and pN+ oral cancer patients treated with SOHND; there was no difference even in regional recurrence outside the surgical field (level IV–V), confirming the rarity of lower neck occult disease. In line with our findings, Kowalski demonstrated 0% of level V involvement/recurrence in N1-2a oral cancer treated with I–III therapeutic neck dissection [24].

Additionally, from another perspective, a few studies have demonstrated an exceedingly low-level V metastasis rate in patients who underwent mRND for pN+ oral cancer, ranging from 0% to 6% in the case of multiple I–IV nodal metastases [48,49]. On the contrary, we encountered only 1.5% level V metastasis following MRND in cN+ patients. Therefore, even in the context of multiple regional metastases, the spread along the nodal echelons has been demonstrated to be constant and predictable.

The evidence from the abundant literature might guide us to safely propose SOHND in cT1-2 N+ oral squamous cell carcinoma [24,35,43,48,49,50,51]. Our current study pointed it out specifically for tongue cancer; moreover, the literature further corroborates our findings that, in the subset of patients affected by T1/2 N1/2, loco-regional control seems to depend on the proper use of adjuvant treatment [28,29] rather than the extension of the neck dissection [24,43,52,53]. Multiple nodal metastases and ENE mainly affect OS, increasing distant failure, not loco-regional control.

Regarding treatment choice in the presence of ENE, our investigation introduces another valuable piece of information: ENE needs to be distinguished from N3b. Our survival analysis in ENE cases has not indicated any benefits in performing mRND compared to SOHND. Recent studies supported these findings and demonstrated how the pN3b category was more predictive of poor prognosis than ENE alone [2,12]. Interestingly, single nodal metastasis with ENE was not significantly associated with a poor prognosis. Once again, a high number of positive nodes [53,54] showed a prognostic weight overwhelming the presence of ENE and other survival indicators. These patients are a minority; ENE is often observed in necks with multiple metastases [2]. We demonstrated that, in cT1-2 N0/+ tongue cancer, SOHND is oncologically safe, even in the presence of ENE.

The strength of our work is that we focused on a large and homogeneous cohort of cT1/T2 OTSCC patients treated in a single institution with uniform treatment guidelines, comparing two different treatment strategies. However, the retrospective nature of this study represents its principal limit. We hope that further research will clarify whether a particular subset of patients with multiple lower neck nodal metastasis might benefit or not benefit from mRND, followed by adjuvant treatment. More interesting pathological factors could be studied in such a setting, such as the lymph-node size, the worst pattern of invasion, and the micro- vs. macro-ENE, to justify the need for adjuvant treatment.

## 5. Conclusions

Treating cT1-2 N0/+ tongue cancer with SOHND is oncologically safe and justified by the close-to-zero rate of skip metastasis. Furthermore, ENE and multiple nodes do not benefit from a more extensive neck dissection since it might affect the distant control primarily.

## Figures and Tables

**Figure 1 jpm-12-01535-f001:**
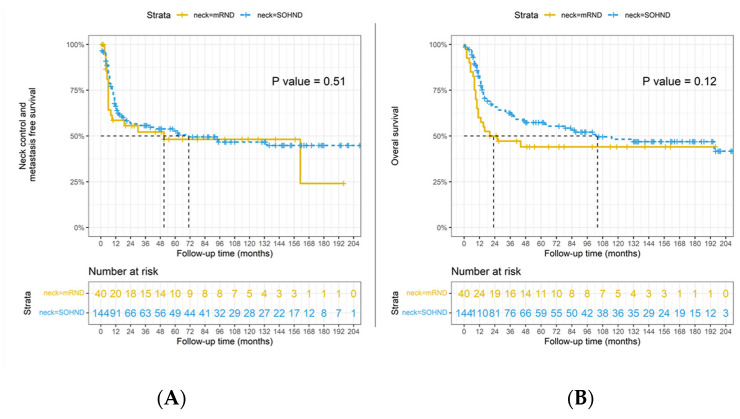
(**A**) Disease-free survival differences in pN+ patients by types of neck dissection (*n* = 184, *p* = 0.51); (**B**) overall survival differences in pN+ patients by types of neck dissection (*p* = 0.12). Dotted line indicated the survival time that half of the patients had no events.

**Figure 2 jpm-12-01535-f002:**
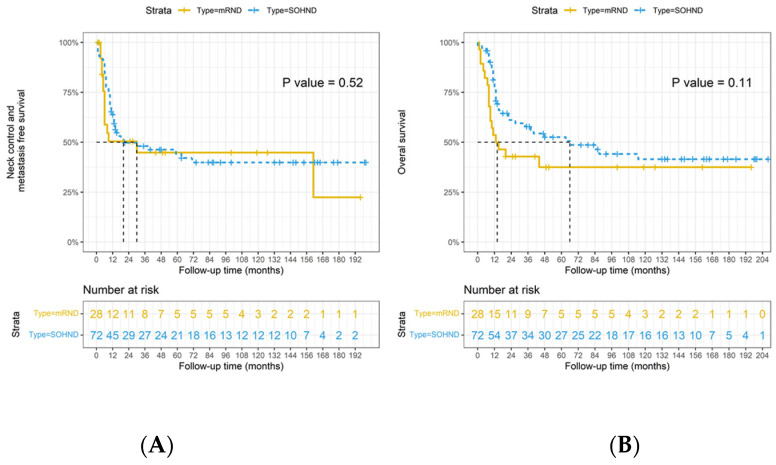
(**A**) Disease-free survival differences in multiple nodal disease by types of neck dissection (*n* =100, *p* = 0.52); (**B**) overall survival differences in multiple nodal disease by types of neck dissection (*p* = 0.11). Neck-control-distant metastases-free survival in multiple nodal disease by types of neck dissection. Dotted line indicated the survival time that half of the patients had no events.

**Figure 3 jpm-12-01535-f003:**
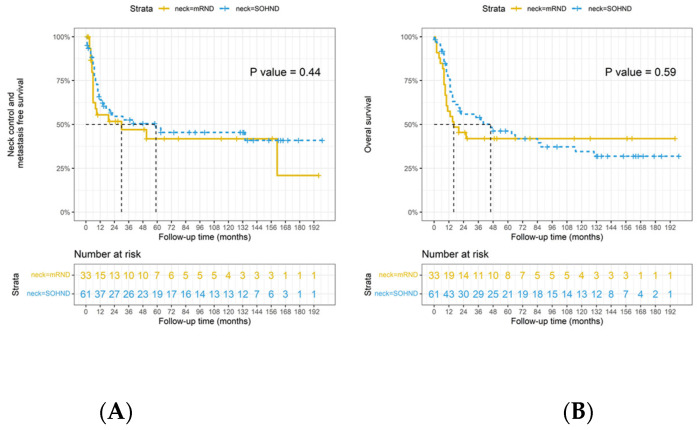
(**A**) Disease-free survival differences in extranodal extension+ (ENE+) by types of neck dissection (*n* = 94, *p* = 0.44); (**B**) overall survival differences in extranodal extension+ (ENE+) by types of neck dissection (*p* = 0.59). Neck-control-distant metastases-free survival in extranodal extension+ (ENE+) by types of neck dissection. Dotted line indicated the survival time that half of the patients had no events.

**Table 1 jpm-12-01535-t001:** Patient’s clinical data and pathological characteristics.

Characteristics		Total	(%)	pN0	(%)	pN+	(%)	*p*
cT stage	T1	256	45.3	215	56.4	41	22.3	<0.001
	T2	309	54.7	151	43.6	107	77.7	
cN stage	N0	381	67.4	-	-	-	-	-
	N1	84	14.9	-	-	-	-	-
	N2b	90	15.9	-	-	-	-	-
	N2c	10	1.8	-	-	-	-	-
pT stage	T1	175	31.0	158	41.	17	9.2	<0.001
	T2	199	35.2	139	36.5	60	32.6	
	T3	191	33.8	84	22.0	107	58.2	
pN stage	N0	381	67.4	-	-	-	-	-
	N1	59	10.4	-	-	-	-	-
	N2b	30	5.3	-	-	-	-	-
	N2c	1	0.2	-	-	-	-	-
	N3b	94	16.6	-	-	-	-	-
Grading	Well	177	31.3	141	37.0	36	19.6	<0.001
	Moderate	331	58.6	215	56.4	116	63.0	
	Poor	57	10.1	25	6.6	32	17.4	
PNI	No	388	68.7	290	76.1	98	53.3	<0.001
	Yes	177	31.3	91	23.9	86	46.7	
VI	No	-	-	-	-	-	-	-
	Yes	-	-	-	-	-	-	-
LI	No	541	95.8	377	99.0	164	89.1	<0.001
	yes	24	4.2	4	1.0	20	10.9	

PNI: perineural invasion; VI: vascular invasion; LI: lymphatic invasion.

**Table 2 jpm-12-01535-t002:** Relationship between pathologic nodal status and type of ND in different AJCC editions.

	AJCC 7th Ed. pN Status	SOHND (%)	mRND (%)	AJCC 8th Ed. pN Status	SOHND (%)	mRND (%)	*p*
pN0	381 (67.4)	357 (71.3)	24 (37.5)	381 (67.4)	357 (71.3)	24 (37.5)	<0.001
pN1	84 (14.9)	72 (14.4)	12 (18.8)	59 (10.4)	55 (11.0)	4 (6.2)	
pN2a				25 (4.4)	17 (3.4)	8 (12.5)	
pN2b	90 (15.9)	66 (13.2)	24 (37.5)	30 (5.3)	27 (5.4)	3 (4.7)	
pN2c	10 (1.8)	6 (1.2)	4 (6.3)	1 (0.2)	1 (0.2)	0 (0.0)	
pN3b				69 (12.2)	44 (8.8)	25 (36.2)	
Total		501 (100.0)	64 (100.0)		501 (100.0)	64 (100.0)	

SOHND: supraomohyoid neck dissection; mRND: modified radical neck dissection.

**Table 3 jpm-12-01535-t003:** Relationship between extranodal extension (ENE) and type of neck dissection in pathologic multiple nodal disease.

	All	SOHND (%)	mRND (%)	*p*
ENE−	31	28 (38.90)	3 (10.70)	<0.001
ENE+	69	44 (61.10)	25 (89.30)	
Total	100	72	28	

SOHND: supraomohyoid neck dissection; mRND: modified radical neck dissection; ENE: extranodal extension.

**Table 4 jpm-12-01535-t004:** Multivariate Cox regression analysis for factors influencing disease-free survival and overall survival.

			DFS					OS
Variables	HR (95% C.I.)	*p*	HR (95% C.I.)	*p*	HR (95% C.I.)	*p*	HR (95% C.I.)	*p*
Age								
<50 years old	Ref	0.037	Ref		Ref			
≥50 years old	0.735 (0.550–0.982)		0.737 (0.548–0.991)	0.043	0.984 (0.715–1.354)	0.921		
Primary tumor status								
pT1	Ref		Ref	-	Ref		Ref	-
pT2	1.489 (1.018–2.179)	0.040	1.144 (0.766–1.707)	0.511	2.862 (1.696–4.829)	<0.001	2.245 (1.310–3.848)	0.003
pT3	2.261 (1.569–3.258)	<0.001	1.417 (0.933–2.151)	0.102	5.032 (3.052–8.298)	<0.001	2.910 (1.691–5.007)	<0.001
Lymph node metastasis								
pN (−) ENE(−)	Ref		Ref	-	Ref		Ref	-
pN (+) ENE(−)	2.045(1.420–2.945)	<0.001	1.603 (1.085–2.369)	0.018	2.632(1.732–4.000)	<0.001	2.001 (1.294–3.094)	<0.001
pN (+) ENE(+)	2.743(1.947–3.865)	<0.001	2.034 (1.381–2.995)	<0.001	5.491(3.828–7.876)	<0.001	3.756 (2.510–5.622)	<0.001
Type of neck dissection								
SOHND	Ref		Ref	-	Ref		Ref	-
mRND	1.624(1.094–2.411)	0.016	1.188 (0.783–1.802)	0.418	2.471 (1.673–3.649)	<0.001	1.348 (0.887–2.048)	0.162
Tumor cell differentiation								
Well/Moderate			Ref	-	Ref			
Poor	1.683 (1.113–2.545)	0.014	1.275 (0.824–1.973)	0.275	1.212 (0.722–2.034)	0.468		
PNI								
No	Ref	<0.001	Ref	-	Ref		Ref	
Yes	1.816(1.363–2.419)		1.398 (1.023–1.910)	0.035	1.644 (1.190–2.271)	0.003	1.006 (0.714–1.417)	0.972

SOHND: supraomohyoid neck dissection; mRND: modified radical neck dissection; ENE: extranodal extension; DFS: disease-free survival; OS: overall survival.

## Data Availability

The data was available by request to the correspondence author.

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
