# Peer review of "Adequacy of Disease Control by Supraomohyoid Neck Dissection in cT1/T2 Tongue Cancer"

_jpm, 2022, doi:10.3390/jpm12091535_

Round 1
Reviewer 1 Report
A brilliant and well presented work. A very good job indeed.
But presenting the subcategory of N0 early oral cancer with separate statistical analysis is suggested.
Best regards
Author Response
1. Thanks for the reviewer’s comments.
2. The subcategory of N0 was used as a reference category when compared with occult metastasis groups (N1, N2b etc..) as shown in Table 4. We thus did not present N0 as another group in our article..
Reviewer 2 Report
too many abbreviations which make reading less fluent
Author Response
1. Thanks for the reviewer’s comment.
2. Most of the abbreviations were commonly used in the head and neck surgical oncology. Some of the abbreviations were used to make the article less lengthy.
3. We thoroughly reviewed our article, we are glad to modify our tables and figures if needed.
Reviewer 3 Report
You have done an interesting job. It is not new, but it is worth periodically publishing new papers providing an exhaustive analysis of the data, as you have done.
I support the publication of your paper, personally it gives me interesting information
Author Response
1. Thanks for the reviewer’s comment.